# Population characteristics, PrEP eligibility, and trust in family planning providers among women accessing public family planning clinics in Kenya

David Mukasa[1]*, John Kinuthia[1,2], Daniel Matemo[2], Jennifer Morton[1], Cynthia Wandera[2], Ugochinyere V. Ukah[3,4], Kenneth K. Mugwanya[1,4], for the FP Plus Project Team

1 Department of Global Health, University of Washington, Seattle, Washington, United States of America, 2 Research & Programs, Kenyatta National Hospital, Nairobi, Kenya, 3 Department of Medicine, McGill University, Montréal, Quebec, Canada, 4 Department of Epidemiology, University of Washington, Seattle, Washington, United States of America

* mukasa@uw.edu

## Abstract

Integrating pre-exposure prophylaxis delivery into family planning (FP) clinics may reach women at elevated HIV risk. We evaluated population characteristics, assessed and perceived HIV risk, and preferences of women accessing real-world FP clinics. Between July 26, 2021, and July 07, 2024, we conducted a cross-sectional study nested within a pragmatic trial of PrEP delivery integrated in twelve real-world FP clinics in Kisumu Kenya (Clinical Trials.Gov: NCT04666792). Quantitative exit surveys were administered to women on randomly selected days to characterize service satisfaction, preferences, behavioral and HIV risk. HIV risk and PrEP eligibility were assessed using Kenya PrEP guidelines. Overall, 1801 participants were interviewed, representing 9.9% of women without HIV accessing the twelve clinics during the 24-month study period. Median age (IQR) was 27.0 (23.2-32.0) years, 37.0% were ≤24 years, and 77.8% were married. Nearly all women (97.5%) visited clinics for FP services; 22.5% used injectable, 19.7% oral pills, and 20.4% implants. Most women (79.1%) reported condomless sex at last sex, but two-thirds were unaware of the HIV status of the person they last had sex with. One-third (30.6%, 551/1801) were assessed to have elevated HIV risk and PrEP eligible, but 72.2% (1300/1801) reported low self-perceived HIV risk and declined PrEP. Most women (78.5%) were not using any HIV prevention strategy, including 88% of those at elevated HIV risk. Knowledge about daily oral PrEP was high (69.5%) but low for injectable PrEP (8.2%). Most women preferred long-acting PrEP, with 70.9% interested in injectables and 39.8% in monthly oral PrEP. Trust in FP providers was high (98.6%), and FP clinics were the preferred access point for HIV prevention care (81.3%). Although women at elevated HIV risk frequently visited FP clinics, 55.3% (305/551) reported low self-perceived risk and low oral PrEP uptake; FP clinics were the preferred and trusted platform for HIV prevention care.

**Data availability statement:** The full original dataset and statistical analysis plan can be accessed by academic researchers by contacting the International Clinical Research Center at the University of Washington (icrc@uw.edu). Data will be shared without investigator support, after approval of a proposal, with a signed data access agreement, for research purposes.

**Funding:** This study was supported by the National Institute of Mental Health of the US National Institutes of Health in the form of a grant awarded to KKM (R01 MH123267 and R00 MH118134). The specific roles of this author are articulated in the 'author contributions' section. The funders had no role in study design, data collection and analysis, decision to publish, or preparation of the manuscript.

**Competing interests:** K.K.M has received research support from Gilead paid to his institution. All authors declare no competing interests.

## Introduction

In HIV high burden settings, many women concerned about avoiding or postponing pregnancy are also at elevated risk for HIV [1]. Modern contraceptive methods protect against unintended pregnancies but, with the exception of condoms, no contraceptive method provides protection against HIV or other sexually transmitted infections (STIs). Indeed, in a large study of HIV acquisition risk among eastern and southern Africa women using depot medroxyprogesterone acetate-intramuscular, the copper intrauterine device or a levonorgestrel implant (the ECHO Study) [2], both HIV (an average nearly 4%) and STI (107.9 per 100 women-years) incidence were alarmingly high during follow up. Higher rates of HIV infection were reported for women under 25 years irrespective of the contraceptive method (3.52 to 4.66 per 100-woman years) [2]. Following these results multiple stakeholders including the World Health Organization have issued a 'Call to Action' to integrate sexual and reproductive health and rights with HIV prevention services, particularly in places where women seek contraception [3].

In many settings in Sub Saharan Africa, family planning (FP) clinics provide broad coverage for women in their reproductive years. For many sexually active women, particularly those from underserved communities, FP clinics offer a range of services, including contraception, pregnancy testing, and HIV counseling and testing, playing a significant role in enabling women to make informed choices about their sexual and reproductive lives. For example, in Kenya, more than 65% of sexually active unmarried women use a modern contraceptive and a substantial proportion (69%) access it through public health FP settings [4]. Thus, anchoring HIV prevention services including provision of PrEP services in FP care settings may offer a tremendous opportunity to reach many women who could benefit from PrEP. Although many stakeholders have called to leverage FP clinics [5], there is a disconnect between policy and actual practice and implementation at country, regional, and facility level [6].

Understanding the characteristics and profile of women who utilize these clinics is essential for tailoring services to meet their specific needs, including access to and provision of HIV prevention services and overall reproductive health outcomes. We conducted a large cross-sectional study to describe the socio-demographic, reproductive health, and HIV risk profiles of women accessing public sector FP clinics to provide insights for program planning.

## Methods and materials

### Ethics statement

This project was approved by the Kenyatta National Hospital-University of Nairobi Ethics Research Committee (P446/08/2020) and the University of Washington Human Subjects Division (STUDY00009583) as a minimal risk study that enabled abstraction of programmatic data and administration of client exit cross-sectional survey. Thus, individual written consent was not required for programmatic services and quality improvement including the exit cross-sectional survey, but oral consenting

was obtained before the exit questionnaire was administered. Consistent with routine services and quality improvement, documentation of oral consent for exit survey was waived by the IRB but the oral consent process was based on an IRB-approved oral consenting guide.

## Study design

Between July 26, 2021, and July 07, 2024, we conducted a cross-sectional study nested within a large pragmatic trial to integrate PrEP delivery in public sector FP clinics in Kenya (the FP Plus project, ClinicalTrials.gov: NCT04666792). The study details and parent protocol are described elsewhere [7]. Briefly, the FP Plus Project was a pragmatic stepped wedge cluster-randomized project to integrate systematic screening and offer of oral PrEP to women accessing twelve primary care FP clinics in Kenya. The intervention package included training of providers and technical support to build the capacity of public sector FP providers to deliver PrEP. The training was facility-based, used the Kenyan Ministry of Health case-based interactive training curriculum, and delivered over a 2-weeks period before each clinic transitioned to the intervention period. Refresher sessions were conducted every three to six months to account for staff turnover. Technical assistance consisted of ongoing coaching and mentoring of healthcare workers in family planning clinics on PrEP delivery. Technical advisors were study-dedicated nurses with training and experience in oral PrEP delivery. They visited each clinic once a week for the first 3 months and then once a month for 9 months until each clinic had received the intervention package for 12 months. All core components of service delivery, including screening for HIV risk and PrEP eligibility, HIV testing, dispensing, adherence and risk reduction counseling, and provision of PrEP refills and contraception services, were delivered by public sector FP clinic staff as part of standard-of-care services with no additional direct resources provided by the project. Within the program, we used cross-sectional quantitative exit surveys administered on random days and clinics by trained study staff to consecutive women as they exited care across 12 clinics. The overall goal of the exit survey was to characterize the profile of women accessing family planning clinics and their experiences, including their HIV risk behaviors, type and quality of services received, and satisfaction with services. The sequence for the specific random days and clinic where the exit surveys would be conducted was generated by Kenya-based study data manager every 3 months without any restriction or stratification. At each clinic, the survey was administered after the clinic had transitioned from usual care to intervention implementation period.

## Study setting, clinic selection, and population

The study was conducted in Kisumu County, Kenya, a region with a population of approximately 1.15 million people and an HIV prevalence of 15.6% among adult population and up to 22% among women [8]. Twelve FP clinics participating in the programmatic project were selected through a joint decision process including study investigators and the MOH and Kisumu County Health officials based on high number of women accessing care, willingness to participate, ability to capture geographic and demographic representation of women accessing FP services and breadth among clinic staff. The program was set up for all sexually active HIV-uninfected women of reproductive age (15–49 years) receiving care at participating clinics. The broad eligibility criteria were designed to reflect the programmatic nature of the primary work – to operate effectively, efficiently, and ethically in public health sector settings. Specific eligibility for cross-sectional survey included seeking care at the clinic on the selected day of the survey at participating FP clinics and willingness to provide oral consent to participate in the exit survey. On selected days, quantitative exit surveys were administered to consecutive women on random select days as they existed the FP clinics.

## Data collection and measurements

We used quantitative exit surveys administered to consecutive women exiting FP clinics to obtain detailed information on characteristics of women and their experiences accessing FP clinics. Structured questionnaires assessed

themes on demographics, trust in FP providers, reasons and frequency of visits to FP clinics, current contraception method use, preferred contraceptives and level of satisfaction, sexual behaviors, HIV risk and risk perception. In addition, the questionnaires were used to collect data on knowledge and access to HIV prevention services including screening for PrEP, PrEP use status, and reasons for not using PrEP if eligible, awareness of different PrEP modalities, preferences, and anticipated concerns about emerging PrEP options. Specifically, access to FP Services was assessed through structured questions on overall satisfaction with FP clinic experience, reasons for visiting FP clinics, number of times they had visited FP clinic in the last 12 months, current contraception method, satisfaction with current contraception method, preferred contraception method. HIV risk constructs were assessed using questions on self-reported HIV testing history, perceived HIV risk, number and type of sexual partners, frequency of condomless sex, alcohol consumption before sex, HIV status of the main partner and the partner with whom the participant last had sex. PrEP eligibility was determined according to the Kenya PrEP guidelines based on behavioral criteria that have been associated with risk of HIV infection. The characteristics defined by the Kenya MOH to indicate elevated risk for HIV and PrEP eligibility include a self-report of any of the following factors in the previous six months: inconsistent or no condom use, having a sex partner(s) at high risk and whose HIV status is unknown, engaging in transactional sex, ongoing intimate partner or gender-based violence, recent sexually transmitted infection, recurrent use of post-exposure prophylaxis, recurrent sex under the influence of alcohol or recreational drugs, injection drug use with shared needles and/or syringes. PrEP awareness and access to HIV Prevention Services were assessed through using structured questions on trust in FP staff to provide adequate HIV prevention care, whether they were screened or not for HIV risk and PrEP eligibility, current method used for HIV prevention, satisfaction with their current HIV prevention methods option, whether they discussed any HIV prevention methods, preferred delivery location to receive HIV prevention care in general, and if it was acceptable to them to receive PrEP through FP clinics. Additional themes assessed were whether they were on or offered PrEP, and if eligible but declined PrEP, reasons for declining/discontinuing PrEP, awareness, and preference of different PrEP options.

## Sample size and statistical analysis

The sample size was calculated to achieve precision estimates for the frequency of key behavior and HIV risk factors to characterize the profile and PrEP eligibility of women accessing public FP clinics. We calculated the sample size using a precision-based equation and stratified by age [9,10] taking into consideration a design effect, with its assumptions. Assuming the proportion ($p$) of women expected to visit clinics to access FP services in Kenya ($p = 0.6$) based on a previous study [11], the minimum proportion of stratification per age group set at 0.5, precision set at 0.10 and design effect of 2, the estimated minimum adjusted sample size was 1,639 women to be interviewed. A total of 1801 women were enrolled to account for potential 10% non-response rate.

Descriptive statistics were used to summarize sociodemographic, access to FP services, preferred contraceptive methods, HIV risk perception, HIV prevention methods, knowledge of PrEP, PrEP use status, and sexual behaviors. Categorical variables were summarized as frequencies and percentages and non-normal continuous variables as medians and interquartile ranges (quartiles). The Chi Square and Fisher Exact Tests were used to assess the differences in distribution of categorical variables stratified by age or PrEP eligibility, as appropriate. Statistical analysis was performed using SAS version 9.4 (SAS Inc., Cary, NC, USA). P values <0.05 were assumed to represent statistical significance in distribution of characteristics stratified by age categories or PrEP eligibility.

## Results

### Participant demographic characteristics

Overall, 18 399 women had their first clinic encounter during the intervention period in the parent study. 1801 of 18 399 women (9.78%) were enrolled and interviewed in the exit cross-sectional survey (Fig 1). The median

PLOS Global Public Health

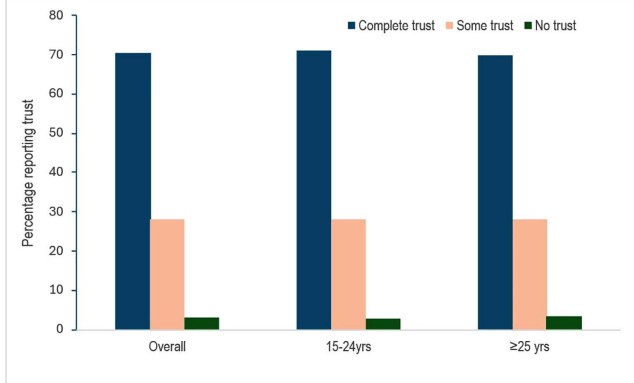

**Fig 1. Overall and age stratified proportions of trusting of family planning providers by women.**

age (IQR) was 27.0 (23.2-32.0) years and 37.0% (665/1,797) were ≤24 years. In total, 77.8% (1401/1801) were married. Nearly two-thirds (59.9% (1079/1801)) had attained secondary school education level and 52.0% (937/1801) had some form of employment. A total of 43.1% (777/1801) had no personal income, including 66.9% (445/665) among women <25 years. Overall, 49.3% (887/1801), 34.6% (392/1801), and 17.1% (306/1801) of the participants were from peri-urban, urban, and rural clinics, respectively (Table 1 and Table A in S1 Text and Table B in S1 Text).

## Access to family planning services, trust in providers, and satisfaction with FP Clinic experience

Overall, nearly all women (97.5%) mentioned that the primary reason for visiting the clinic was to seek contraception services and about 15% mentioned HIV prevention services including HIV testing. At the interview encounter, most women (68.2%) were on a modern contraception method, including 22.5% on injectable, 19.7% on oral pills, and 20.4% on implant contraception methods. Among women interviewed, 52.8% (952/1801) had two-three FP clinic visits over the previous 12 months; 35.9% had their first visit to the FP clinic in the previous 12 months (44.0% for women <25 years versus 31.2% for ≥25 years). Three-fourths of the participants (76.6% (1379/1801)) reported that the FP healthcare provider discussed HIV prevention with them during their visit, including information on HIV testing, partner HIV testing and status, and PrEP. Nearly all women (>98.6% overall, 70.6% complete trust, 28.0% some trust) expressed trust and confidence in their FP healthcare providers to give adequate care and information about HIV prevention services, with similar frequency among women <25 years vs those ≥25 years (Fig 1). Similarly, most women (93.6%, 1686/1801) reported to be satisfied with their overall FP clinic experience after the visit (Table 1).

## Sexual behaviors, HIV risk characteristics, and PrEP eligibility

The sexual behaviors and HIV risk characteristics of survey participants are presented in Table 2. Overall, 418 of 1801 (23.2%) participants reported having more than one current sexual partner (32.5% for <25 years versus 17.9% for ≥25 years) and 79.8% (1437/1801) reported having at least one main sexual partner. 1263 of 1801 (70.1%) participants did not know the HIV status of their primary main sexual partner, 79.1% (1424/1801) reported condomless sex at their last sex encounter, but 64.0% (1152/1801) did not know the HIV status of the person they last had sex with (67.8% for <25 years versus 60.8% for ≥25 years). About one-third (30.6%, 551/1801) of women (46.1% for <25 years versus 53.9% for ≥25 years) were assessed to be eligible for PrEP based on characteristics defined by the Kenya PrEP guidelines.

**Table 1. Demographic Characteristics of Women Accessing Family Planning Services.**

| Covariates | ∏Overall (N = 1,801) | 15-24 years (N = 665) | 25-49 years (N = 1,132) |
|---|---|---|---|
| | n/N (%) or Median (IQR) | n/N (%) or Median (IQR) | n/N (%) or Median (IQR) |
| **Demographics** | | | |
| **Age in years, Median (Q1-Q3)** | 27.0 (23.2-32.0) | 22.3 (20.4-23.6) | 30.1 (27.6- 34.2) |
| **Age Category** | | | |
| 15-19 | 117/1801 (6.5) | 117/665 (17.6) | – |
| 20-24 | 548/1801 (30.5) | 548/665 (82.4) | – |
| 25-34 | 865/1801 (48.0) | – | 865/1132 (76.4) |
| ≥ 35 | 267/1801 (14.8) | – | 267/1132 (23.6) |
| Missing | 4/1801 (0.2) _ | – | – |
| **Marital status** | | | |
| Married | 1401/1801 (77.8) | 437/665 (65.7) | 963/1132 (85.1) |
| Not Married | 373/1801 (20.7) | 219/665 (32.9) | 154/1132 (13.6) |
| Not Applicable/Unknown | 27/1801 (1.5) | 9/665 (1.4) | 15/1132 (1.3) |
| **Education** | | | |
| Primary and below | 297/1801 (16.5) | 64/665 (9.6) | 233/1132 (20.6) |
| Secondary Education | 1079/1801 (59.9) | 469/665 (70.5) | 609/1132 (53.8) |
| Attended post-secondary school | 398/1801 (22.1) | 123/665 (18.5) | 275/1132 (24.3) |
| Unknown | 27/1801 (1.5) | 9/665 (1.4) | 15/1132 (1.3) |
| **Occupation** | | | |
| Employed (Formal/Informal) | 937/1801 (52.0) | 190/665 (28.6) | 746/1132 (65.9) |
| Unemployed | 395/1801 (21.9) | 198/665 (29.8) | 197/1132 (17.4) |
| Student Status | 228/1801 (12.7) | 202/665 (30.4) | 26/1132 (2.3) |
| Household Wife/Child Care | 213/1801 (11.8) | 66/665 (9.9) | 147/1132 (13.0) |
| Missing | 28/1801 (1.6) | 9/665 (1.3) | 16/1132 (1.4) |
| **Personal Income** | | | |
| No Income | 777/1801 (43.1) | 445/665 (66.9) | 332/1132 (29.3) |
| 1–5,000 Ksh | 205/1801 (11.4) | 65/665 (9.8) | 140/1132 (12.4) |
| 5,001–10,000 Ksh | 320/1801 (17.8) | 78/665 (11.7) | 242/1132 (21.4) |
| > 10,000 ksh | 277/1801 (15.4) | 34/665 (5.1) | 242/1132 (21.4) |
| Declined Response/Unknown | 222/1801 (12.3) | 43/665 (6.5) | 176/1132 (15.5) |
| **Location of the Clinics** | | | |
| Rural | 306/1801 (16.9) | 106/665 (15.9) | 199/1132 (17.6) |
| Peri-Urban | 887/1801 (49.3) | 346/665 (52.0) | 541/1132 (47.8) |
| Urban | 608/1801 (33.8) | 213/665 (32.0) | 392/1132 (34.6) |
| **Reason for visiting FP clinic*** | | | |
| Family planning | 1756/1801 (97.5) | 656/665 (98.6) | 1099/1132 (97.1) |
| HIV testing or Prevention/ STI testing or Treatment | 264/1801 (14.7) | 102/665 (15.3) | 162/1132 (14.3) |
| Cervical cancer screening | 126/1801 (7.0) | 31/665 (4.7) | 95/1132 (8.4) |
| Others | 276/1801 (15.3) | 101/665 (15.2) | 175/1132 (15.5) |
| **Number of times visited FP clinic in last 12 months** | | | |
| First visit today | 646/1801 (35.9) | 293/665 (44.0) | 353/1132 (31.2) |
| 2–3 visits | 952/1801 (52.8) | 343/665 (51.6) | 608/1132 (53.7) |
| 4 or more visits | 200/1801 (11.1) | 29/665 (4.4) | 171/1132 (15.1) |
| Unknown | 3/1801 (0.2) | 0/665 (0.0) | 0/1132 (0.0) |

*(Continued)*

**Table 1.** (Continued)

| Covariates | ∏Overall (N = 1,801) | 15-24 years (N = 665) | 25-49 years (N = 1,132) |
|---|---|---|---|
| | n/N (%) or Median (IQR) | n/N (%) or Median (IQR) | n/N (%) or Median (IQR) |
| **Access to Contraception** | | | |
| **Any Modern Contraception** | | | |
| Yes | 1228/1801 (68.2) | 460/665 (69.2) | 768/1132 (67.9) |
| None | 94/1801 (5.2) | 28/665 (4.2) | 66/1132 (5.8) |
| Missing | 479/1801 (26.6) | 177/665(26.6) | 298/1132 (26.3) |
| **Current FP Method*** | | | |
| Implant | 368/1801 (20.4) | 162/665 (24.4) | 206/1132 (18.2) |
| Injectable | 405/1801 (22.5) | 142/665 (21.4) | 263/1132 (23.2) |
| OCP | 354/1801 (19.7) | 133/665 (20.0) | 221 (19.5) |
| IUCD | 75/1801 (4.2) | 15/665 (2.3) | 60/1132 (5.3) |
| Condoms and Sterilization | 27/1801 (1.5) | 9/665 (1.4) | 18/1132 (1.6) |
| None | 94/1801 (5.2) | 28/665 (4.2) | 66/1132 (5.8) |
| **Received Preferred FP method** | | | |
| Yes | 1061/1801 (58.9) | 387/665 (58.2) | 673/1132 (59.5) |
| No/ Not sure | 155/1801 (8.6) | 62/665 (9.3) | 93/1132 (8.2) |
| Not Applicable/ didn't receive a method | 585/1801 (32.5) | 216/665 (32.5) | 366/1132 (32.3) |
| **What they would have wanted to receive*** | | | |
| Implant | 66/740 (8.9) | 30/278(10.8) | 36/459 (7.8) |
| Injectable | 74/740 (10.0) | 29/278 (10.4) | 45/459 (9.8) |
| OCP | 16/740 (2.2) | 4/278 (1.4) | 12/459 (2.6) |
| IUCD | 6/740 (0.8) | 2/278 (0.7) | 4/459 (0.9) |
| Others | 13/740 (1.8) | 3/278 (1.1) | 10/459 (2.2) |
| None | 36/740 (4.9) | 10/278 (3.6) | 26/459 (5.7) |
| Missing | 529/740 (71.5) | 200/278 (71.9) | 326/459 (71.0) |
| **Overall Clinical Experience/Satisfaction** | | | |
| Very Satisfied/Somewhat satisfied | 1686/1801 (93.6) | 626/665 (94.1) | 1059/1132 (93.6) |
| Totally Dissatisfied/Somewhat dissatisfied | 112/1801 (6.2) | 39/665 (5.9) | 73/1132 (6.4) |
| Unknown | 3/1801 (0.2) | 0/665 (0.0) | 0/1132 (0.00) |

*We included 4 participants with missing age data. The age stratified analysis comprises 1,797 participants instead of 1801 participants: *Multiple choice selection; OCP: Oral Contraceptive Pills; IUCD: Intrauterine Contraceptive Device.

### HIV risk perception, PrEP awareness, uptake, and preference

Details on HIV risk perception, awareness about PrEP and preference for different PrEP modalities are summarized in Table 3. Overall, more than 95.4% (1,717/1801) of women reported testing for HIV at least once in the previous 12 months, including 50.2% (904/1801) who had tested within 3 months of the interview. Nearly three-quarters (72.2%, 1300/1801)) of participants reported to have zero to very low self-perceived risk for HIV, including more than half (55.4%, 305/551) of those assessed to be at elevated HIV risk and deemed eligible for PrEP. Most participants (78.5%, 1413/1801) reported not to be on any HIV prevention strategy at the time, including 88.2% (486/551) of those assessed by the provider to be at increased HIV risk and eligible for PrEP. Despite many women not using any HIV prevention method, two-thirds of women (64.3%, 1158/1801) reported to be very satisfied/somewhat satisfied with their current HIV prevention choice, including 72.4% (399/551) of those assessed to

**Table 2. Sexual behavior and HIV risk Characteristics.**

| Covariates | ∏Overall (N = 1,801) | 15-24 years (n = 665) | 25-49 years (n = 1,132) |
|---|---|---|---|
| | n/N (%) or Median (IQR) | n/N (%) or Median (IQR) | n/N (%) or Median (IQR) |
| **Sexual Risk Behaviors** | | | |
| **Alcohol Consumption Before Sex** | | | |
| Yes | 264/1801 (14.6) | 137/665 (20.6) | 127/1132 (11.2) |
| No | 1426/1801 (79.2) | 478/665 (71.9) | 94/1132 (83.7) |
| Unknown | 111/1801 (6.2) | 50/665 (7.5) | 58/1132 (5.1) |
| **Condomless Sex with Partner of Unknown HIV Status** | | | |
| Yes | 1186/1801 (65.9) | 442/665 (66.5) | 744/1132 (65.7) |
| No | 588 (32.6) | 214/665 (32.2) | 373/1132 (33.0) |
| Unknown | 27 (1.5) | 9/665 (1.3) | 15/1132 (1.3) |
| **Recent STI Diagnosis** | | | |
| Yes | 165/1801 (9.2) | 64/665 (9.6) | 101/1132 (8.9) |
| No | 1609/1801 (89.3) | 592/665 (89.0) | 1016/1132 (89.8) |
| Unknown | 27 (1.5) | 9/665 (1.4) | 15/1132 (1.3) |
| **Shared Needles** | | | |
| Yes | 8/1801 (0.4) | 3/665 (0.5) | 5/1132 (0.4) |
| No | 1765/1801 (98.0) | 652/665 (98.0) | 1112/1132 (98.2) |
| Unknown | 28/1801 (1.6) | 10/665 (1.5) | 5/1132 (0.4) |
| **Transactional Sex** | | | |
| Yes | 254/1801 (14.1) | 150/665 (22.6) | 104/1132 (9.2) |
| No | 1519/1801 (84.3) | 505/665 (75.9) | 1012/1132 (89.5) |
| Unknown | 28/1801 (1.6) | 10/665 (1.5) | 15/1132 (1.3) |
| **Sexually Assaulted** | | | |
| Yes | 182/1801 (10.1) | 72/665 (10.8) | 110/1132 (9.7) |
| No | 1591/1801 (88.3) | 584/665 (87.8) | 1005/1132 (88.8) |
| Unknown | 28/1801 (1.6) | 9/665 (1.4) | 16/1132 (1.4) |
| **PEP Use** | | | |
| Yes | 25/1801 (1.4) | 9/665 (1.34) | 16/1132 (1.4) |
| No | 1748/1801 (97.1) | 646/665 (97.1) | 1100/1132 (97.3) |
| Unknown | 28/1801 (1.5) | 10/665 (1.5) | 15/1132 (1.3) |
| **PrEP Eligibility** | | | |
| Yes | 551/1801 (30.6) | 254/665 (38.2) | 297/1132 (26.3) |
| No | 1250/1801 (69.4) | 411/665 (61.8) | 834/1132 (73.7) |
| **Number of Current Sexual Partners** | | | |
| None | 84/1801 (4.7) | 48/665 (7.2) | 36/1132 (3.2) |
| One Partner | 1272/1801 (70.6) | 392/665 (59.0) | 878/1132 (77.6) |
| >1 Partner | 418/1801 (23.2) | 216/665 (32.5) | 202/1132 (17.9) |
| Unknown | 27/1801 (1.5) | 9/665 (1.4) | 15/1132 (1.3) |
| **Number of Main Sexual Partners** | | | |
| None | 218/1801 (12.1) | 106/665 (15.9) | 112/1132 (9.9) |
| One Partner | 1437/1801 (79.8) | 487/665 (73.3) | 948/1132 (83.8) |
| Two Partners | 16/1801 (0.9) | 8/665 (1.2) | 8/1132 (0.7) |
| Unknown | 130/1801 (7.2) | 64/665 (9.6) | 6/1132 (5.6) |
| **Number of Casual Sexual Partners** | | | |
| None | 1168/1801 (64.9) | 350/665 (52.6) | 816/1132 (72.1) |

*(Continued)*

**Table 2.** (Continued)

| Covariates | ∏Overall (N = 1,801) | 15-24 years (n = 665) | 25-49 years (n = 1,132) |
|---|---|---|---|
| | n/N (%) or Median (IQR) | n/N (%) or Median (IQR) | n/N (%) or Median (IQR) |
| One Partner | 312/1801 (17.3) | 151/665 (22.7) | 161/1132 (14.2) |
| >1 Partner | 193/1801 (10.7) | 102/665 (15.3) | 91/1132 (8.1) |
| Decline Response/Unknown | 128/1801 (7.1) | 62/665 (9.3) | 63/1132 (5.6) |
| **Having Sex with New Sex Partners in the past 3 months** | | | |
| Yes | 234/1801 (12.99) | 128/665 (19.25) | 106/1132 (9.36) |
| No | 1540/1801 (85.51) | 528/665 (79.40) | 1011/1132 (89.31) |
| Unknown | 27/1801 (1.50) | 9/665 (1.35) | 15/1132 (1.33) |
| **Frequency of condomless sex with casual sexual Partners** | | | |
| Always | 68/505 (13.4) | 38/253 (11.8) | 30/252 (9.3) |
| Often | 121/505 (24.0) | 51/253 (15.8) | 71/252 (22.1) |
| Sometimes | 158/505 (31.3) | 91/253 (28.3) | 67/252 (20.8) |
| Rarely | 112/505 (22.2) | 53/253 (16.5) | 59/252 (18.3) |
| Never | 44/505 (8.7) | 19/253 (5.9) | 26/252 (8.1) |
| Unknown | 2/501 (0.4) | 1/253 (0.4) | 1/252 (0.4) |
| **Condon use at last sex act** | | | |
| Yes | 202/1801 (11.2) | 88/665 (13.2) | 114/1132 (10.1) |
| No | 1424/1801 (79.1) | 495/665 (74.4) | 928/1132 (82.0) |
| Part of the time | 146/1801 (8.1) | 73/665 (11.1) | 73/1132 (6.4) |
| Unknown | 29 (1.6) | 9/665 (1.3) | 17/1132 (1.5) |
| **HIV Status of main sexual partner** | | | |
| Negative | 536/1801 (29.8) | 187/665 (28.1) | 348/1132 (30.7) |
| Positive | 2/1801 (0.1) | 0/665 (0.0) | 2/1132 (0.2) |
| Unknown | 1263/1801 (70.1) | 478/665 (71.9) | 782/1132 (69.1)) |
| **HIV status of partner last had sex with** | | | |
| HIV negative | 635/1801 (35.2) | 210/665 (32.0) | 424/1132 (38.0) |
| HIV positive | 14/1801 (0.8) | 1/665 (0.2) | 13/1132 (1.2) |
| Unknown | 1152/1801 (64.0) | 445/665 (67.8) | 678/1132 (60.8) |

*A total of 4 participants did not have data on age and were not included in the age stratified analysis; *Multiple choice selection; PMTCT; Prevention of Mother to Child Transmission.

be at elevated risk for HIV. More than half (55.5%, 307/551) of participants who were provider-assessed to be at elevated risk for HIV declined when offered PrEP or had discontinued at time of the interview. Overall, the most frequently reported reasons for declining oral PrEP were related to low perceived risk for HIV (20.8%), needing time to decide (20.3%), need to first consult their partner or family (9.8%), concern about family/friends finding out and related stigma (10.3%), and concern about pills burden (13.3%). Overall, 1242 of 1801 (68.9%) of women were knowledgeable and aware about daily oral PrEP pill for HIV prevention prior to clinic visit, but only 9.9% (178/1801) had ever heard about injectable PrEP (13.8% for <25 years versus 8.2% for ≥25 years). Nevertheless, most women expressed preference for long-acting PrEP options, with 70.9% of participants expressing preference for injectable PrEP form, 39.8% stating interest in monthly oral pill, and 36.1% stated interest in implant form of PrEP. Among women expressing interested in injectable PrEP, a majority (64.8%) preferred using a semi-annual injection option.

**Table 3.** HIV prevention characteristics Stratified by PrEP Eligibility among women Accessing Family Planning.

| Covariates | ∏Overall (N = 1,801) | Eligible for PrEP (n = 551) | Not Eligible for PrEP (n = 1250) |
|---|---|---|---|
| | n/N (%) or Median (IQR)) | n/N (%) or Median (IQR) | n/N (%) or Median (IQR) |
| **Age (Continuous), n = 1797** | 27.0 (23.2-32.0) | 25.7 (22.1-30.8) | 27.5 (23.8-32.0) * |
| **Age (Categorized)** | | | |
| 15–24 years | 665/1801 (36.9) | 254/551 (46.1) | 411/1250 (32.9) |
| 25–34 years | 865/1801 (48.0) | 221/551 (40.1) | 644/1250 (51.5) |
| ≥35 years | 267/1801 (14.9) | 76/551 (13.8) | 191/1250 (15.3) |
| Missing | 4/1801 (0.2) | 0/551 (0.00) | 4/1250 (0.3) |
| **Location of the Clinics** | | | |
| Rural | 306/1801 (16.9) | 66/551 (12.0) | 240/1250 (19.2) |
| Peri-Urban | 887/1801 (49.3) | 293/551 (53.2) | 594/1250 (47.5) |
| Urban | 608/1801 (33.8) | 192/551 (34.8) | 416/1250 (33.3) |
| **When last tested for HIV** | | | |
| <3 months | 904/1801 (50.2) | 273/551 (49.5) | 631/1250 (50.5) |
| 3–6 months | 495/1801 (27.5) | 156/551 (28.3) | 339/1250 (27.1) |
| 7–12 months | 318/1801 (17.7) | 100/551 (18.2) | 218/1250 (17.4) |
| >12 months/ Never tested before/Declined response | 84/1801 (4.6) | 22/551 (4.0) | 62/1250 (5.0) |
| **Perceived risk** | | | |
| Zero (No Risk) | 743/1801 (41.3) | 115/551 (20.9) | 628/1250 (50.2) |
| Low | 557/1801 (30.9) | 190/551 (34.5) | 367/1250 (29.4) |
| Moderate | 380/1801 (21.1) | 174/551 (31.6)) | 206/1250 (16.5) |
| High/ Very High | 118/1801 (6.5) | 72/551 (13.0) | 46/1250 (3.7) |
| Unknown | 3/1801 (0.2) | 0/551 (0.00) | 3/1250 (0.2) |
| **Current method used for HIV prevention*** | | | |
| Nothing | 1413/1801 (78.5) | 486/551 (88.2) | 927/1250 (74.2) |
| Consistent condom use | 75/1801 (4.2) | 36/551 (6.5) | 39/1250 (3.1) |
| PrEP | 34/1801 (1.9) | 18/551 (3.3) | 16/1250 (1.3) |
| HIV+ partner is on ART | 2/1801 (0.1) | 1/551 (0.2) | 1/1250 (0.1) |
| N/A, I don't feel at risk for HIV | 277/1801 (15.4) | 10/551 (1.8) | 267/1250 (21.4) |
| **Satisfaction with their current HIV prevention status** | | | |
| Very satisfied/Somewhat satisfied | 1158/1801 (64.3) | 399/551 (72.4) | 759/1250 (60.7) |
| Totally dissatisfied/Somewhat dissatisfied | 112/1801 (6.2) | 37/551 (6.7) | 75/1250 (6.0) |
| Neutral/ Not Applicable | 531/1801 (29.5) | 115/551 (20.9) | 416/1250 (33.3) |
| **Prevention access** | | | |
| **HIV prevention method discussed with provider*** | | | |
| None | 273/1801 (15.2) | 71/551 (12.8) | 202/1250 (16.2) |
| HIV testing/ Partner HIV testing & status | 1379/1801 (76.6) | 451/551 (81.9) | 928/1250 (74.2) |
| STI testing and counselling/treatment of STIs | 683/1801 (37.9) | 189/551 (34.3) | 494/1250 (39.5) |
| PEP/ Assessed or counseled about oral PrEP | 1354/180 (75.2) | 434/551 (78.8) | 920/1250 (73.6) |
| PMTCT | 37/180 (2.1) | 7/551 (1.3) | 30/1250 (2.4) |
| Condoms/Others | 338/1801 (18.8) | 94/551 (17.1) | 244/1250 (19.5) |
| **What they would have preferred*** | | | |
| Nothing | 1215/1801 (67.5) | 353/551 (64.1) | 862/1250 (69.0) |
| HIV testing and counselling/ Partner HIV testing & status | 272/1801 (15.1) | 85/551 (15.4) | 187/1250 (15.0) |
| STI testing and counselling/treatment of STIs | 166/1801 (9.2) | 32/551 (5.8) | 134/1250 (10.7) |
| PEP/ Assessed or counseled about oral PrEP | 323/1801 (17.9) | 131/551 (23.8) | 192/551 (15.4) |

*(Continued)*

| Covariates | ⊓Overall (N = 1,801) | Eligible for PrEP (n = 551) | Not Eligible for PrEP (n = 1250) |
|---|---|---|---|
| | n/N (%) or Median (IQR)) | n/N (%) or Median (IQR) | n/N (%) or Median (IQR) |
| PMTCT | 65/1801 (3.6) | 16/551 (2.9) | 49/1250 (3.9) |
| Condoms/Others | 77/1801 (4.28) | 39/551 (7.1) | 38/1250 (3.0) |
| **Preferred location to receive HIV prevention services*** | | | |
| FP clinics | 1464/1801 (81.3) | 432/551 (78.4) | 1032/1250 (82.6) |
| Antenatal clinics | 164/1801 (9.1) | 44/551 (8.0) | 120/1250 (9.6) |
| HIV Clinics | 199/1801 (11.1) | 74/551 (13.4) | 125/1250 (10.0) |
| STI clinics | 57/1801 (3.2) | 15/551 (2.7) | 42/1250 (3.4) |
| Outpatient department | 126/1801 (7.0) | 38/551 (6.9) | 88/1250 (7.0) |
| Community Pharmacy | 136/1801 (7.6) | 43/551 (7.8) | 93/ 1250 (7.4) |
| Direct from pharmacy within health facilities | 285/1801 (15.8) | 105/551 (19.1) | 180/1250 (14.4) |
| Drop-in centers/Others | 359/1801 (19.9) | 142/551 (25.8) | 217/1250 (17.4) |
| **PrEP use** | | | |
| **Initiation status** | | | |
| On PrEP | 82/1801 (4.6) | 31/551 (5.6) | 51/1250 (4.1) |
| Declined PrEP/Discontinued PrEP | 821/1801 (45.6) | 307/551 (55.7) | 514/1250 (41.1) |
| Desired PrEP but clinic did not provide PrEP/referral | 111/1801 (6.2) | 62/551 (11.3) | 49/1250 (3.9) |
| Does not need PrEP (not at risk) | 246/1801 (13.7) | 31/551 (5.6) | 215/1250 (17.2) |
| Not applicable | 541/1801 (30.0) | 120/551 (21.8) | 421/1250 (33.6) |
| **Reasons not PrEP*** | | | |
| Lack of information/Appropriateness/Efficacy/Trust | 180/1719 (10.5) | 75/520 (14.4) | 105/1199 (8.8) |
| Does not feel at risk for HIV | 358/1719 (20.8) | 48/520 (9.2) | 310/1199 (25.9) |
| Too many pills to take everyday | 228/1719 (13.3) | 106/520 (20.4) | 122/1199 (10.2) |
| Side effects | 215/1719 (12.5) | 100/520 (19.2) | 115/1199 (9.6) |
| Family and friend Concerns | 177/1719 (10.3) | 91/520 (17.5) | 86/1199 (7.2) |
| Need to consult partner/Family and Friends | 168/1719 (9.8) | 64/520 (12.3) | 104/1199 (8.7) |
| Need time make Decision | 366/1719 (20.3) | 164/520 (31.5) | 202/1199 (16.9) |
| Commuting Time/Waiting Time at FP Clinics | 64/1719(3.7) | 29/520 (5.6) | 35/1199 (2.9) |
| Staff/clinic was not able to provide PrEP/Others | 178/1719 (10.4) | 95/520 (7.9) | 83/1199 (16.0) |
| **Knowledge about oral PrEP** | | | |
| Yes | 1242/1801 (68.9) | 373/551 (67.7) | 869/1250 (69.5) |
| No | 383/1801 (21.3) | 114/551 (20.7) | 269/1250 (21.5) |
| Not sure | 173/1801 (9.6) | 64/551 (11.6) | 109/1250 (8.7) |
| Unknown/Missing | 3/1801 (0.2) | 0/551 (0.0) | 3/1250 (0.3) |
| **Knowledge about injectable PrEP** | | | |
| Yes | 178/1801 (9.9) | 76/551 (13.8) | 103/1250 (8.2) |
| No | 1094/1801 (60.7) | 360/551 (65.3) | 734/1250 (58.7) |
| Unknown | 529/1801 (29.4) | 115/551 (20.9) | 414/1250 (33.1) |
| **Concern about Injectable PrEP use*** | | | |
| No concerns/fears at all | 456/1801 (25.3) | 185/551 (33.6) | 271/1250 (21.7) |
| Safety or side effects | 417/1801 (23.2) | 133/551 (24.1) | 284/1250 (22.7) |
| Pain from Injections/Fears or dislike needles | 275/1801 (15.3) | 105/551 (19.1) | 170/1250 (13.6) |
| Weigt gain | 40/1801 (2.2) | 6/551 (1.1) | 34/1250 (2.7) |
| Having to return for injections every two months | 98/1801 (5.4) | 44/551 (8.0) | 54/1250 (4.3) |
| Lack of information/Interest in PrEP/Others | 516/1801 (28.7) | 157/551 (28.5) | 359/1250 (28.7) |

*(Continued)*

**Table 3.** (Continued)

| Covariates | ∏Overall (N = 1,801) | Eligible for PrEP (n = 551) | Not Eligible for PrEP (n = 1250) |
|---|---|---|---|
| | n/N (%) or Median (IQR)) | n/N (%) or Median (IQR) | n/N (%) or Median (IQR) |
| **Preference for different PrEP forms *** | | | |
| Daily pill | 61/1801 (3.4) | 13/551 (2.4) | 48/1250 (3.8) |
| Monthly pill | 716/1801 (39.8) | 224/551 (40.7) | 492/1250 (39.4) |
| Intravaginal ring | 87/1801 (4.8) | 24/551 (4.4) | 63/1250 (5.04) |
| Injectable | 1276/1801 (70.9) | 396/551 (71.9) | 880/1250 (70.4) |
| Implant | 650/1801 (36.1) | 235/551 (42.7) | 415/1250 (33.2) |
| Others | 22/1801 (1.2) | 5/551 (0.9) | 17/1250 (1.4) |
| **Preferred Frequency of Injectable PrEP** | | | |
| A bimonthly injection (every two months) | 93/1276 (7.3) | 26/457 (5.7) | 66/818 (8.0) |
| A semi-annual injection (every six months) | 827/1276 (64.8) | 307/457 (67.2) | 520/818 (63.6) |
| Unknown/Missing | 356/1276 (27.9) | 124/457 (27.1) | 232/818 (28.4) |

*A total of 4 participants did not have data on age; PMTCT, Prevention of Mother to Child Transmission; *Multiple choice selection.

## Discussion

In this large cross-sectional study, most women attending public sector FP clinics had low awareness of their sexual partner HIV status, and majority expressed low self-perceived HIV risk despite most of them being assessed to be at elevated risk for HIV. Women expressed high preference for FP clinics as their primary location and overwhelming trust in FP healthcare providers to serve their HIV prevention service needs. These findings show that FP clinics have tremendous potential to serve as a one-stop client-centered key access point to reach women for wholistic sexual and reproductive care services, including provision of PrEP services for HIV prevention.

Public FP clinics are highly accessed by sexually active women in many African settings [11]. While as expected the primary reason for visiting the FP clinic was for contraceptive needs, a notable proportion also sought HIV prevention services, such as testing, as their primary reason for attending, indicating an opportunity to leverage this existing platform for expanded HIV prevention efforts. Importantly, most women interviewed expressed high satisfaction and preference for FP clinics as their primary location for HIV prevention services (81.3%) and reported overwhelming trust in FP healthcare providers as source of care and information about HIV prevention services. Very often health providers in this setting are not given the due recognition they deserve as we tend to only focus on the negative reported attributes. This often ignores the context in which these providers work, including an overburdened health system and high client volumes. The high level of trust and satisfaction towards FP service providers and FP clinics as an access venue supports the promise and acceptability of integrated FP and HIV prevention service care for women able to prefer and are able to access facility-based care.

Identifying characteristics of women accessing public sector FP clinics offers an opportunity to understand how to target and expand comprehensive healthcare services. Many women accessing FP clinics in this study had behavioral factors linked to elevated risk for HIV, including multiple concurrent partners and inconsistent condom use [12,13]. Notably, although most women had tested for HIV at least once in the previous 12-months, a substantial proportion (>70.1%) were unaware of the HIV status of their primary sexual partner and more than half of those classified as eligible for PrEP expressed low self-perceived risk for HIV. In many studies, self-perceived HIV risk has been associated with high levels of PrEP uptake and persistence [14,15]. Thus, the disconnect between provider-assessed HIV risk and self-perceived vulnerability for HIV points to the need for enhanced strategies that help women better assess their own.

 

Despite high awareness of oral PrEP (68.9%), there was limited uptake of daily oral PrEP at the clinic when first offered, mainly due to low perceived risk, the need for more time to consult partner/family, potential stigma, and concern about pill burden. These findings are consistent with results from a recent systematic review assessing barriers to PrEP uptake in multiple countries including Kenya and South Africa which highlighted the same obstacles to PrEP uptake and utilization [16]. In contrast, awareness of injectable PrEP was low, which was in harmony with another study conducted in Uganda that also reported a very low proportion (3.9%) of women who knew injectable PrEP [17]. Despite low awareness about injectable PrEP, there was a clear preference for long-acting PrEP options, with a significant interest in both injectable PrEP and less frequent oral PrEP. This study was conducted in the context with only oral PrEP as the available standard of care PrEP option. Although long-acting injectable cabotegravir and dapivirine vaginal ring are already approved in Kenya, they are yet available as standard of care in public health clinics. Lenacapavir received regulatory approval in Kenya in early 2026; however, widespread availability as standard of care will depend on national rollout, guideline inclusion, and implementation timelines. A recent demonstration project in Uganda and Kenya that offered a person-centered intervention with choice of delivery point and PrEP products, including long-acting injectable cabotegravir improved HIV prevention coverage by more than 50% compared to standard of care [18]. Taken together, this demonstrates the tremendous potential to expand coverage and the reach for new users if choice and access to these emerging multiple options with improved availability and equitable access.

This study has strengths and limitations. The study strengths include a large sample size of women across variety of clinics with high volume and heterogenous social, economic, and geographical settings in Kenya. Our sample of 1801 women were drawn from the general population of women who visited the 12 FP clinics participating in the large programmatic cluster randomized trial during the 24 months. There are also limitations to our study. First, this was a cross-sectional study prone to the inherent flaws of this exploratory design. Secondly, women were recruited consecutively based on self-selection and the findings are based self-reported data. Thirdly, although survey items were derived from psychometrically developed instruments validated for use in this setting, the survey did not administer the full original instruments, which may limit assessment of psychometric properties and comparability with prior studies. Importantly, we have previously employed this tool and its components in several clinical and implementation studies within the same population, such as the Partners Scale-Up in HIV care clinics and the PrIYA program in MCH/FP, which strengthens our confidence in its content validity. Fourthly, for this cross-sectional survey, however, we did not examine stigma and partner influence in detail, and therefore it would be speculative to draw conclusions regarding these factors. Importantly, within the larger parent study, we conducted complementary qualitative interviews with women, which will provide deeper insights into these contextually important themes of stigma and partner influence. Despite these limitations, this large cross-sectional study provides important empirical evidence on the characteristics of women seeking FP services in Western Kenya and the uniqueness of FP clinics as a one-stop access point for comprehensive women care, including PrEP for HIV prevention.

## Conclusions

In summary, in this large cross-sectional study among women accessing high-volume real-world FP clinics in Kenya, we have shown that FP clinics represent a preferred and trusted platform for integrating and expanding HIV prevention services for women in Africa settings. The findings show that FP clinics are uniquely positioned to serve as key access points for women who may benefit from comprehensive healthcare services, including PrEP for HIV prevention. Future research should focus on addressing gaps in HIV risk perception, knowledge of sexual partner status, and PrEP awareness with targeted interventions and defining efficient delivery models integrating emerging long-acting PrEP options in real-world settings.

## Supporting information

**S1 Text. Demographic Characteristics of Women Accessing Family Planning Services, stratified by Location.**
(DOCX)

## Acknowledgments

We thank all women who participated in this study and the staff at implementing facilities and the project staff for their motivation and dedication.

## Author contributions

**Conceptualization:** Jennifer Morton, Kenneth K. Mugwanya.

**Formal analysis:** David Mukasa.

**Funding acquisition:** Kenneth K. Mugwanya.

**Investigation:** Jennifer Morton, Kenneth K. Mugwanya.

**Methodology:** David Mukasa, Kenneth K. Mugwanya.

**Project administration:** John Kinuthia, Daniel Matemo, Cynthia Wandera.

**Writing – original draft:** David Mukasa, Kenneth K. Mugwanya.

**Writing – review & editing:** David Mukasa, Kenneth K. Mugwanya, Ugochinyere V. Ukah.

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
