## [Decision Letter · Decision Letter 0]

22 Sep 2025

PGPH-D-25-02465

Population Characteristics, PrEP Eligibility, and Trust in Family Planning Providers among Women accessing Public Family Planning Clinics in Kenya

Dear Dr. MUKASA,

Thank you for submitting your manuscript to PLOS Global Public Health. After careful consideration, we feel that it has merit but does not fully meet PLOS Global Public Health’s publication criteria as it currently stands. Therefore, we invite you to submit a revised version of the manuscript that addresses the points raised during the review process.

We look forward to receiving your revised manuscript.

Kind regards,

Andrew Kazibwe, MBChB

Academic Editor

Journal Requirements:

1. In the ethics statement in the Methods, you have specified that verbal consent was obtained. Please provide additional details regarding how this consent was documented and witnessed, and state whether this was approved by the IRB.

2. Please include a complete copy of PLOS’ questionnaire on inclusivity in global research in your revised manuscript. Our policy for research in this area aims to improve transparency in the reporting of research performed outside of researchers’ own country or community. The policy applies to researchers who have travelled to a different country to conduct research, research with Indigenous populations or their lands, and research on cultural artefacts. The questionnaire can also be requested at the journal’s discretion for any other submissions, even if these conditions are not met. Please find more information on the policy and a link to download a blank copy of the questionnaire here: https://journals.plos.org/globalpublichealth/s/best-practices-in-research-reporting. Please upload a completed version of your questionnaire as Supporting Information when you resubmit your manuscript.

Additional Editor Comments (if provided):

Editor Query:

The corresponding author should submit a signed cover letter.

I request the authors to provide a table summarising number of participants by study site

Check grammar in lines 43, 98, 128, 145, and 280

Define technical support as used in line 101 - how was this different from training? What nature of training was given to the providers - classroom/online/on-site, skills or knowledge-based? How long was the training?

Who administered the exit interviews? Were the interviewers part of the service providers?

Provide tool for HIV risk assessment

How were days for the interviews selected? How was randomness ensured in selection of days for data collection? How was this expected to influence study results? Would the results have been different if the data collection was done on purposively selected days such as FP clinic days?

In table 1 age categorisations are redundant as row headers

Similarly, in tables 2 and 3, the age and PrEP eligibility disaggregation as column headers make interpretation of p-values cumbersome.

Reviewer #1:

Well written manuscript overall. Large sample size enabling to draw conclusions. However, there are a few clarifications needed.

Lines 71-72 It will be helpful to give the statistics in the text, not only the citation, for ease of reading.

Line 145 " inconsistent OR condom use. Probably this is a typo for OF

Line 150 Was the questionnaire to assess the trust in staff a standard one from literature or otherwise? Was it piloted? Was this based on a documented theoretical framework?

Line 155 ...whether they on or offered PrEP...Is the verb "were" missing here?

Line 267 "unwanted pregnancy". Unplanned pregnancy is the preferred term, as a pregnancy occurring without adequate protection is expected. Moreover, women living with HIV have healthy babies with adequate prophylaxis. The sentence in the manuscript implies otherwise.

Line 289 It is of interest to know if long-lasting PrEP is available and subsidised in the country

Lines 297, 300 Could the authors clarify the terms used: randomized trial (implying comparison group) or cross-sectional study with random sampling methods. These are to distinctive study designs.

In-text: It would be interesting to know the reasons clients were dissatisfied with their clinic experience and had a lack of confidence in FP clinic staff. What theoretical framework informed measurement of patient satisfaction?

Reviewer #2:

Novelty & Context

• Congratulations to the research team for highlighting the novelty of real-world family planning (FP) clinic data from Kenya. This contribution is valuable for understanding HIV prevention opportunities in routine service delivery settings.

Study Period

• Line 35 states the study period as June 2021 – April 2024, whereas line 95 lists July 2021 – July 2024. Please align these dates and specify exact months to ensure clarity and reproducibility.

Data Presentation & Quantification

• Line 49: Self-perceived risk and reasons for PrEP decline are summarized qualitatively but not fully quantified, limiting interpretive power. Consider including proportions or counts to strengthen the impact of this section.

• Lines 51, 54, 55, 201, 207: Several metrics lack precision. Examples include “>69%” for oral PrEP knowledge, “>98” for trust in providers without a stated scale, and “>80%” for FP clinics as the preferred access point. Please use exact percentages (or clearly defined approximate ranges) and specify measurement scales (e.g., Likert) to improve statistical validity and reproducibility.

Terminology Consistency

• Minor: Lines 41 and 191 use “findings” and “results” interchangeably. For coherence, choose one term and apply consistently throughout.

Denominators & Missing Data

• Lines 217, 232, 235, 236 report denominators that vary substantially (e.g., condom use n=505 in line 217 vs. n=5512 — larger than the total sample size of 1801). Please clarify how missing data were handled (e.g., exclusion, imputation) and use consistent denominators or justify differences.

Subgroup Analyses

• The results do not explore urban–rural or clinic-type differences despite the diversity of sites. Consider adding a subgroup or stratified analysis, even as a supplementary table, to provide contextual nuance and inform targeted interventions.

Discussion Enhancements

• Comparative references to Uganda and South Africa are relevant but dated (pre-2024). Consider citing recent 2024–2025 studies on long-acting PrEP (e.g., cabotegravir trials) to highlight the contemporary relevance of the Kenya findings.

• Ethical implications, particularly stigma in partner consultation, are underexplored. Adding a paragraph on stigma mitigation strategies and ethical safeguards would strengthen this section and align with human rights-based approaches.

Conclusion & Forward-Looking Recommendations

• While the conclusion effectively summarizes findings and emphasizes the importance of FP clinics in HIV prevention, it largely repeats the discussion. Strengthen this section by adding forward-looking, actionable recommendations—such as scalable provider training models, differentiated service delivery approaches, or digital tools to support adherence—while emphasizing ethical and equitable scale-up for underserved women.

Reviewers' comments:

Reviewer's Responses to Questions

**Comments to the Author**

1. Does this manuscript meet PLOS Global Public Health’s publication criteria?

Reviewer #1: Yes

Reviewer #2: Yes

2. Has the statistical analysis been performed appropriately and rigorously?

Reviewer #1: I don't know

Reviewer #2: Yes

3. Have the authors made all data underlying the findings in their manuscript fully available (please refer to the Data Availability Statement at the start of the manuscript PDF file)?

Reviewer #1: Yes

Reviewer #2: Yes

4. Is the manuscript presented in an intelligible fashion and written in standard English?

Reviewer #1: Yes

Reviewer #2: Yes

Reviewer #1: Well written manuscript overall. Large sample size enabling to draw conclusions. However, there are a few clarifications needed.

Lines 71-72 It will be helpful to give the statistics in the text, not only the citation, for ease of reading.

Line 145 " inconsistent OR condom use. Probably this is a typo for OF

Line 150 Was the questionnaire to assess the trust in staff a standard one from literature or otherwise? Was it piloted?

Line 155 ...whether they on or offered PrEP...Is the verb "were" missing here?

Line 267 "unwanted pregnancy". Unplanned pregnancy is the prefered term, as a pregnancy occuring without adequate protection is expected. Moreover, women living with HIV have healthy babies with adequate prophylaxis. The sentence in the manuscript implies otherwise.

Line 289 It is of interest to know if long-lasting PrEP is available and subsidised in the country

Lines 297, 300 Could the authors clarify the terms used: randomized trial (implying comparison group) or cross-sectional study with random sampling methods. These are to distinctive study designs.

In-text: It would be interesting to know the reasons clients were disatisfied with their clinic experience and had a lack of confidence in FP clinic staff.

Reviewer #2: Novelty & Context

• Congratulations to the research team for highlighting the novelty of real-world family planning (FP) clinic data from Kenya. This contribution is valuable for understanding HIV prevention opportunities in routine service delivery settings.

Study Period

• Line 35 states the study period as June 2021 – April 2024, whereas line 95 lists July 2021 – July 2024. Please align these dates and specify exact months to ensure clarity and reproducibility.

Data Presentation & Quantification

• Line 49: Self-perceived risk and reasons for PrEP decline are summarized qualitatively but not fully quantified, limiting interpretive power. Consider including proportions or counts to strengthen the impact of this section.

• Lines 51, 54, 55, 201, 207: Several metrics lack precision. Examples include “>69%” for oral PrEP knowledge, “>98” for trust in providers without a stated scale, and “>80%” for FP clinics as the preferred access point. Please use exact percentages (or clearly defined approximate ranges) and specify measurement scales (e.g., Likert) to improve statistical validity and reproducibility.

Terminology Consistency

• Minor: Lines 41 and 191 use “findings” and “results” interchangeably. For coherence, choose one term and apply consistently throughout.

Denominators & Missing Data

• Lines 217, 232, 235, 236 report denominators that vary substantially (e.g., condom use n=505 in line 217 vs. n=5512 — larger than the total sample size of 1801). Please clarify how missing data were handled (e.g., exclusion, imputation) and use consistent denominators or justify differences.

Subgroup Analyses

• The results do not explore urban–rural or clinic-type differences despite the diversity of sites. Consider adding a subgroup or stratified analysis, even as a supplementary table, to provide contextual nuance and inform targeted interventions.

Discussion Enhancements

• Comparative references to Uganda and South Africa are relevant but dated (pre-2024). Consider citing recent 2024–2025 studies on long-acting PrEP (e.g., cabotegravir trials) to highlight the contemporary relevance of the Kenya findings.

• Ethical implications, particularly stigma in partner consultation, are underexplored. Adding a paragraph on stigma mitigation strategies and ethical safeguards would strengthen this section and align with human rights-based approaches.

Conclusion & Forward-Looking Recommendations

• While the conclusion effectively summarizes findings and emphasizes the importance of FP clinics in HIV prevention, it largely repeats the discussion. Strengthen this section by adding forward-looking, actionable recommendations—such as scalable provider training models, differentiated service delivery approaches, or digital tools to support adherence—while emphasizing ethical and equitable scale-up for underserved women.

**Do you want your identity to be public for this peer review?** For information about this choice, including consent withdrawal, please see our Privacy Policy

Reviewer #1: **Yes:** maria a afadapa

Reviewer #2: No

---

## [Decision Letter · Decision Letter 1]

6 Jan 2026

PGPH-D-25-02465R1

Population Characteristics, PrEP Eligibility, and Trust in Family Planning Providers among Women accessing Public Family Planning Clinics in Kenya

Dear Dr. MUKASA,

Thank you for submitting your manuscript to PLOS Global Public Health. After careful consideration, we feel that it has merit but does not fully meet PLOS Global Public Health’s publication criteria as it currently stands. Therefore, we invite you to submit a revised version of the manuscript that addresses the points raised during the review process.

Thank you for addressing the reviewers' comments in the current revision. Please address the additional minor comments and return the revised version as soon as possible.

We look forward to receiving your revised manuscript.

Kind regards,

Andrew Kazibwe, MBChB, MMED

Academic Editor

Journal Requirements:

Additional Editor Comments (if provided):

This is a well written manuscript with a current topic

Line 118 Population of Kisumu County?

Line 190 … “informed” consent…Also specify how <18 gave consent and why it is allowed in this context

Line 202 Women enrolled 1801, sample size determination was 1639. Make a note of why? E. g., attrition rate consideration?

Line 278 Did you probe for the fear of reprisal that could influence response?

Line 284 …opportunity to “know” how…

Line 304 Remove “but”

Line 305 What is “ap”?

Line 317, 319 ‘Secondly’ twice

Line 319-320 Explain how this is a limitation

Table 1 Were there any specific reasons for dissatisfaction in overall clinical experience?

Reviewers' comments:

Reviewer's Responses to Questions

**Comments to the Author**

Reviewer #1: (No Response)

Reviewer #2: All comments have been addressed

publication criteria?

Reviewer #1: Yes

Reviewer #2: Yes

3. Has the statistical analysis been performed appropriately and rigorously?

Reviewer #1: Yes

Reviewer #2: Yes

4. Have the authors made all data underlying the findings in their manuscript fully available (please refer to the Data Availability Statement at the start of the manuscript PDF file)?

Reviewer #1: Yes

Reviewer #2: Yes

5. Is the manuscript presented in an intelligible fashion and written in standard English?

Reviewer #1: Yes

Reviewer #2: Yes

Reviewer #1: This is a well written manuscript with a current topic

Line 118 Population of Kisumu County?

Line 190 … “informed” consent…Also specify how <18 gave consent and why it is allowed in this context

Line 202 Women enrolled 1801, sample size determination was 1639. Make a note of why? E. g., attrition rate consideration?

Line 278 Did you probe for the fear of reprisal that could influence response?

Line 284 …opportunity to “know” how…

Line 304 Remove “but”

Line 305 What is “ap”?

Line 317, 319 ‘Secondly’ twice

Line 319-320 Explain how this is a limitation

Table 1 Were there any specific reasons for dissatisfaction in overall clinical experience?

Reviewer #2: Congratulations for addressing all comments in detail. Please ensure the line spacing from 312 to 331 are in sync with the other sections. Congrats once again to all the authors.

**Do you want your identity to be public for this peer review?** For information about this choice, including consent withdrawal, please see our Privacy Policy

Reviewer #1: **Yes:** Maria Afadapa

Reviewer #2: **Yes:** Richard Noamesi AMENYAH

---

## [Editor Report · Decision Letter 2]

29 Jan 2026

Population Characteristics, PrEP Eligibility, and Trust in Family Planning Providers among Women accessing Public Family Planning Clinics in Kenya

PGPH-D-25-02465R2

Dear Dr MUKASA,

We are pleased to inform you that your manuscript 'Population Characteristics, PrEP Eligibility, and Trust in Family Planning Providers among Women accessing Public Family Planning Clinics in Kenya' has been provisionally accepted for publication in PLOS Global Public Health.

Best regards,

Andrew Kazibwe, MBChB, MMED

Academic Editor